# Comparative genomic analysis of *Methylocystis* sp. MJC1 as a platform strain for polyhydroxybutyrate biosynthesis

Sanzhar Naizabekov[1☯], Seung Woon Hyun[1☯], Jeong-Geol Na[2], Sukhwan Yoon[3], Ok Kyung Lee[1]*, Eun Yeol Lee[1]*

1 Department of Chemical Engineering (BK21 FOUR Integrated Engineering Program), Kyung Hee University, Yongin-si, Gyeonggi-do, Republic of Korea, 2 Department of Chemical and Biomolecular Engineering, Sogang University, Seoul, Republic of Korea, 3 Department of Civil & Environmental Engineering, Korea Advanced Institute of Science & Technology, Daejeon, Republic of Korea

☯ These authors contributed equally to this work.
* eunylee@khu.ac.kr (EYL); okblessyou@khu.ac.kr (OKL)

**Data Availability Statement:** The genome sequence of Methylocystis sp. MJC1 was deposited in DDBJ/ENA/GenBank under accession number CP107558.

## Abstract

Biodegradable polyhydroxybutyrate (PHB) can be produced from methane by some type II methanotroph such as the genus *Methylocystis*. This study presents the comparative genomic analysis of a newly isolated methanotroph, *Methylocystis* sp. MJC1 as a biodegradable PHB-producing platform strain. *Methylocystis* sp. MJC1 accumulates up to 44.5% of PHB based on dry cell weight under nitrogen-limiting conditions. To facilitate its development as a PHB-producing platform strain, the complete genome sequence of *Methylocystis* sp. MJC1 was assembled, functionally annotated, and compared with genomes of other *Methylocystis* species. Phylogenetic analysis has shown that *Methylocystis parvus* to be the closest species to *Methylocystis* sp. MJC1. Genome functional annotation revealed that *Methylocystis* sp. MJC1 contains all major type II methanotroph biochemical pathways such as the serine cycle, EMC pathway, and Krebs cycle. Interestingly, *Methylocystis* sp. MJC1 has both particulate and soluble methane monooxygenases, which are not commonly found among *Methylocystis* species. In addition, this species also possesses most of the RuMP pathway reactions, a characteristic of type I methanotrophs, and all PHB biosynthetic genes. These comparative analysis would open the possibility of future practical applications such as the development of organism-specific genome-scale models and application of metabolic engineering strategies to *Methylocystis* sp. MJC1.

## Introduction

Methane is a potent greenhouse gas with a higher potential to cause global warming than that of carbon dioxide [1]. The methane cycle is a part of the broader carbon cycle, which is a biochemical process that facilitates the flow of carbon in the atmosphere. Wetlands, such as swamps, marches, and bogs, are places where most parts of the methane cycle occur [2]. Methanotrophs are wetland-based microorganisms that consume methane as a carbon source, thus decreasing the extent of methane emission in the atmosphere.

**Funding:** This research was supported by the C1 Gas Refinery Program through the National Research Foundation of Korea (NRF) funded by the Ministry of Science and ICT (2015M3D3A1A01064882 and 2015M3D3A1A01064926). This research was also supported by Basic Science Research Program through the National Research Foundation Korea (NRF) funded by the Ministry of Education (2020R1I1A1A01073467). The funders had no role in study design, data collection and analysis, decision to publish, or preparation of the manuscript.

**Competing interests:** The authors declare that they have no competing interests.

On the other side, methane is an important chemical feedstock because it is a primary component of abundant natural gas and biogas. In the conventional in-direct approaches, the chemical conversion processes of methane into other chemicals require a high amount of energy because of the high activation energy of the carbon-hydrogen bond [3]. Methanotrophs, on the other hand, can produce other substances from methane without requiring such high energy inputs. The maximum carbon conversion efficiency of methane to methanol oxidation using chemical processes has been reported to be less than 50%, while that of methanotrophs was reported to be up to 75% [4].

For this reason, methanotrophs have been spotlighted as platform strains that produce high value-added products from methane. Methanotrophs are traditionally categorized into two groups based on their carbon assimilation pathways: Type I and Type X belong to gamma-proteobacteria, while Type II belongs to alpha-proteobacteria. Type I and X use the ribulose monophosphate (RuMP) cycle to assimilate C1 carbon sources such as formaldehyde, whereas Type II uses the serine cycle to assimilate formate generated by methane oxidation [5].

Type I methanotrophs including *Methylomonas* sp. and *Methylomicrobium* sp. condense formaldehyde to ribulose monophosphate, producing fructose-6-phosphate. This metabolite is integrated into various metabolic pathways such as the Embden-Meyerhof-Parnas, oxidative, and non-oxidative pentose phosphate pathways [6]. In order to produce high-value products such as 2,3-BDO, succinic acid, shinorine, and auxin from methane, native or non-native biosynthetic pathways have been successfully introduced into central carbon metabolism of type I methanotrophs [7–10]. Nonetheless, the low productivity of these target products needs to be addressed.

Type II methanotrophs, such as *Methylocystis* sp. and *Methylosinus* sp., have the ability to accumulate polyhydroxybutyrate (PHB) around 34–70% of dry cell weight through a series of enzymatic reactions involving acetyl-CoA acetyltransferase, acetoacetyl-CoA reductase, and PHA synthase [11–13]. Recently, PHB production by Type I methanotroph *Methylomicrobium alcaliphilum* 20Z has also been reported, but its feasibility of commercialization remains low [14]. Therefore, Type II methanotrophs are considered promising PHB platform strains. In particular, *Methylocystis* sp. has even more potential because of its higher growth rate (up to 0.12–0.16 $h^{-1}$) compared to *Methylosinus* sp. [15]. Despite these promising advantages, only six species of the *Methylocystis* genus have been reported, thus more strain needs to be isolated and characterized to develop PHB-producing platforms [16, 17].

For industrial application of PHB production from methane by methanotrophs, isolation, characterization and genome analysis of many methanotrophic strains are prerequisites for industrial strain development and further metabolic engineering for enhanced PHB productivity. In an attempt to develop a methanotrophic platform, a novel type II methanotrophic strain, *Methylocystis* sp. MJC1, isolated from distinct alpine peat bogs in Ulsan, Korea, was characterized in this study [18]. The genome of *Methylocystis* sp. MJC1 was sequenced, assembled and functionally annotated. We evaluated the potential of *Methylocystis* sp. MJC1 as a platform strain for PHB production. In addition, comparative genomic analysis was performed to assess *Methylocystis* sp. MJC1 in the context of other *Methylocystis* clade strains.

## Materials and methods

### Cultivation of methanotroph

*Methylocystis* sp. MJC1 was cultivated using either NMS medium at pH 6.5 with methane as the carbon source, as described previously [19]. Bacterial pre-cultures were grown in a 180 mL baffled flask in 10 mL methane-supplemented medium (30% v/v) at 30˚C and 230 rpm. Subsequently, large-scale cultivation was carried out by transferring pre-cultures into 50 mL of new medium in a 500 mL baffled flask.

## PHB production by *Methylocystis* sp. MJC1

For PHB production by *Methylocystis* sp. MJC1, gas fermentation was conducted in a 5 L glass-bioreactor containing 3 L of NMS medium. Seed cultures grown in a baffled flask were inoculated at a 10% volume ratio. A gas mixture of 30% $CH_4$ and 70% air was supplied at a flow rate of 0.2 vvm using a mass flow controller. Gas fermentation was conducted at 30°C and the impeller speed was in the range of 300–700 rpm. The pH value was maintained at 6.5 by supplying a 0.1 N HCl solution for NMS medium. The optical density was measured at 600 nm using an UV-visible spectrophotometer.

For PHB analysis, the bacterial cells were grown for 4 days and then harvested by centrifugation. The cell cultures were pelleted and freeze-dried 2 days using the freeze dryer. PHB content was analyzed via GC (Agilent GC-8890 MSD; USA) after methanolysis. Methanolysis was performed using 30–50 mg of lyophilized cell in 2 mL of chloroform and 2 mL of methanol containing 15% sulfuric acid in borosilicate glass tubes with screw caps. The reaction mixtures were incubated at 100°C for 3 h in oil-bath. After cooling, 1 mL of distilled water was added, and the tubes were vortexed for 60s. The lower organic phase was filtered (0.2 μm, PTFE membrane) and analyzed by a GC fitted with an HP5 capillary column and equipped with a flame ionization detector (FID).

## Genome sequencing, assembly, annotation and comparative genomics analysis

The extraction of genomic DNA was carried out with the help of Wizard Genomic DNA Purification Kit (Promega, Madison, WI, USA). The complete genome sequencing of *Methylocystis* sp. MJC1 was performed using Macrogen's PacBio RS II and Illumina HiSeq 2500 sequencing platforms. PacBio reads were used to generate a genome draft, which was then further refined using Illumina reads. The resultant 764 Mb filtered polymerase reads were assembled into three contigs using RS_HGAP.3, a hierarchical genome assembly process. The identification of overlaps at both ends of the sequences revealed that all three contigs had circular structures.

Functional genome annotation was performed using NCBI's PGAP (Prokaryotic Genome Annotation Pipeline) standalone software as it provided maximum possible coverage for genome annotation in comparison with other major genome annotation pipelines [20, 21]. Genomic circular maps were rendered using the CGView server [22].

In addition, *Methylocystis* sp. MJC1's central carbon metabolism pathways were refined with the help of pairwise ortholog analysis using annotated genomes of related strains and an extensive literature review. The related strains referred to were *Methylocystis* sp. B8, *Methylococcus capsulatus* Bath, *Methylomonas* sp. DH-1, and *M. trichosporium* OB3b. The ortholog analysis was run using InParanoid software [23]. Finally, the potential secondary metabolites were identified using antiSMASH [24].

16S rRNA gene analysis was performed using NCBI BLAST and Ezcloud web-server [25, 26]. Multiple sequence alignment and phylogenetic tree construction were carried out using MEGA X standalone software [27]. The genomes and proteomes of *Methylocystis* species strains were obtained from NCBI using the experimental datasets command line interface. Electronic DNA–DNA hybridization (DDH) estimates were calculated using the Genome-to-Genome Distance Calculator [28]. The average amino acid identity (AAI) analysis was performed using CompareM [29]. The average nucleotide identity (ANI) analysis was performed using pyani [30]. Cluster of orthologous groups (COG) was obtained using webMGA server [31]. Pangenome analysis and phylogenetic tree construction for concatenated core genes (core genome) shared by *Methylocystis* strains was performed using BPGA software with default parameters [32].

## Results and discussion

### Feasibility evaluation of *Methylocystis* sp. MJC1 as a PHB-producing platform strain

Batch cultures were performed in a gas bioreactor system to investigate cell growth and PHB production of the *Methylocystis* sp. MJC1. The maximum growth rate of the *Methylocystis* sp. MJC1 in the exponential growth phase was 0.12 h$^{-1}$, which is higher than the previously reported cell growth rate of MJC1 (0.063 h$^{-1}$) cultured in DNMS (dilute nitrate mineral salts) medium with the addition of copper [18]. During 4 days of fermentation, the *Methylocystis* sp. MJC1 showed a rapid growth pattern and PHB accumulation started from the exponential growth phase. As shown in **Fig 1**, after 36 h, the biomass increased rapidly and the PHB content gradually increased. PHB accumulation in the exponential phase was expected to occur due to the lack of a nitrogen source in the medium. The biomass and PHB contents produced in the bioreactor were 7.35 g biomass/L and 3.25 g PHB/L, respectively. To date, there are only six reported names for the genus *Methylocystis*, including *Methylocystis parvus* [33], *Methylocystis rosea* [34], *Methylocystis echinoides* [35], *Methylocystis hirsuta* [36], *Methylocystis heyeri* [37], *and Methylocystis bryophila* [38], and technical reports on PHB production using these strains are very limited. Although it is difficult to make accurate quantitative comparisons under the same conditions, nonetheless, *Methylocystis* sp. MJC1 could be one of the promising candidate as a platform strain for production of PHB based on its higher specific growth rate and high PHB content compared to the available references [39–41]. Thus, we conducted genomic assembly, gene annotation, and comparative genomic analysis of *Methylocystis* sp. MJC1 for further studies on metabolic engineering and genome scale metabolic model development.

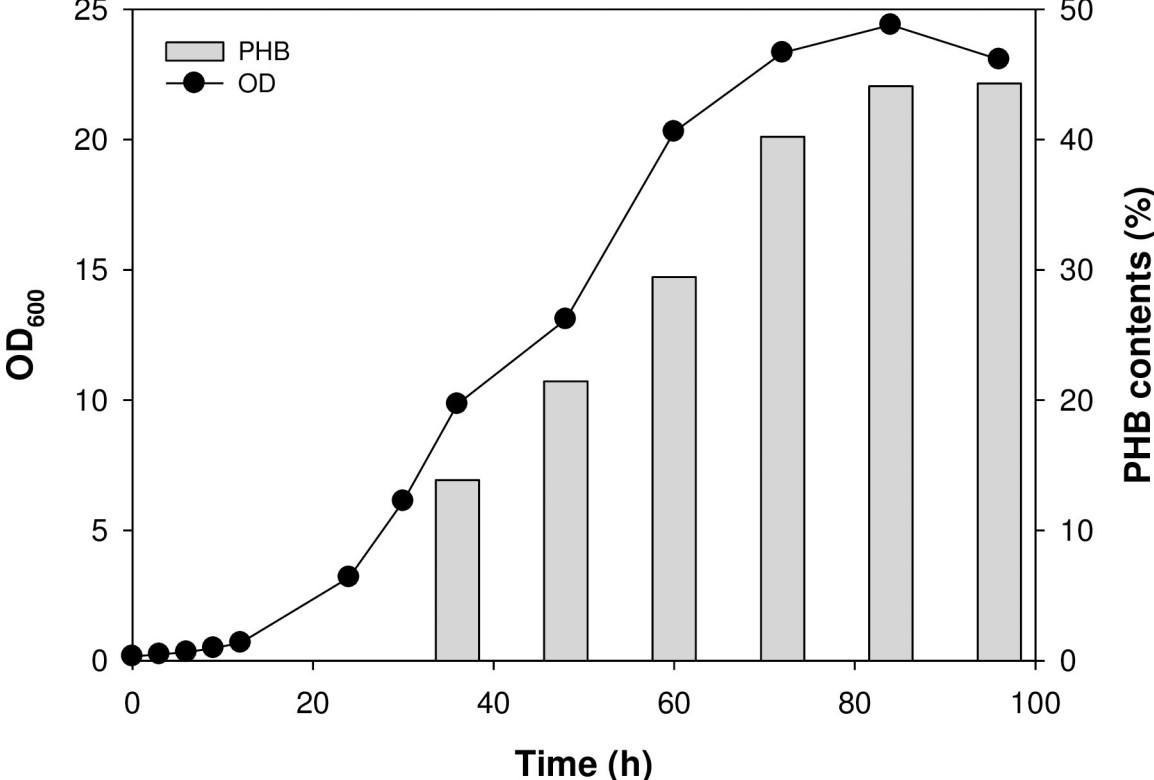

**Fig 1. The growth profile and PHB production of *Methylocystis* sp. MJC1 in bioreactor with NMS medium.**

**Table 1. Comparison of general features of genomes of *Methylocystis* sp. MJC1 and other methanotroph.**

| Feature | *Methylocystis* sp. MJC1 | *Methylocystis* sp. SC2 | *Methylocystis* sp. Rockwell | *Methylosinus trichosporium* OB3b |
|---|---|---|---|---|
| Accession number | CP107558 | HE956757 | AEVM00000000 | ADVE00000000 |
| Assembly level | Complete | Complete | Scaffold | Complete |
| Genome size (bp) | 3,923,488 | 3,773,444 | (4,725,934)* | 4,962,262 |
| G+C content (%) | 62.9 | 63 | 63 | 66 |
| Total no. of CDS | 4,285 | 3,666 | 4,551 | 4,512 |
| Pseudogenes | 134 | 38 | 167 | 110 |
| tRNAs | 53 | 46 | 51 | 47 |
| rRNAs | 3,3,3 (5S, 16S, 23S) | 1, 1, 1 (5S, 16S, 23S) | 2, 2, 2 (5S, 16S, 23S) | 2, 2, 2 (5S, 16S, 23S) |
| ncRNAs | 4 | 4 | 4 | 4 |
| pMMO | + | + | + | + |
| sMMO | + | - | - | + |
| Serine pathway gene | present | present | present | present |
| RuMP pathway gene | partially present | absent | absent | absent |
| PHB synthesis gene | present | present | present | present |
| Plasmid | 2 | 2 | NR** | 3 |
| Reference | This study | [42] | [42] | [42] |

* Total sequence length

** NR-not reported

## Genome features and statistics

The genome of *Methylocystis* sp. MJC1 consists of one chromosome and two plasmids. The assembled genomes has gaps and consists of two linear contigs, whereas both plasmids are complete and circular. The chromosome of 3,923,488 bp and two plasmids of 353,831 (plasmid 1) and 133,722 bp (plasmid 2), with an average GC content of 62.9, 59.7 and 60.2%, respectively (**Table 1** and **Fig 2**).

## Phylogeny and species identification

*Methylocystis* sp. MJC1 has been shown to be phylogenetically closest to *Methylocystis parvus* among *Methylocystis* species strains based on a wide range of bioinformatics analyses,

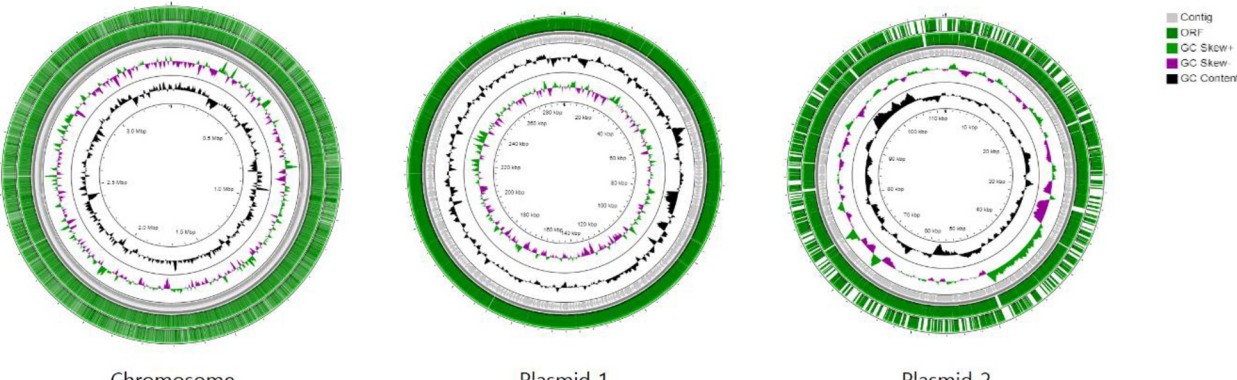

Chromosome Plasmid 1 Plasmid 2

**Fig 2. *Methylocystis* sp. MJC1 contigs in circular representation.** This is a visual representation of the circular map of the MJC1 genome. The genome map consists of three circles. The two outer circles display the open reading frame, while the inner circle represents the GC skew. Positive and negative values of the GC skew are depicted by green and purple, respectively. The GC content is displayed in black. This genome map was generated using the CGView Server.

including 16S rRNA similarity, average amino acid identity (AAI), and average nucleotide identity (ANI) (**S1** and **S2** **Tables**). In addition, the neighbor-joining method algorithm for PmoA phylogenetic tree analysis showed *M. parvus* to be the closest species to *Methylocystis* sp. MJC1 (**Fig 3**). The phylogenetic tree for the concatenated core genes also confirmed

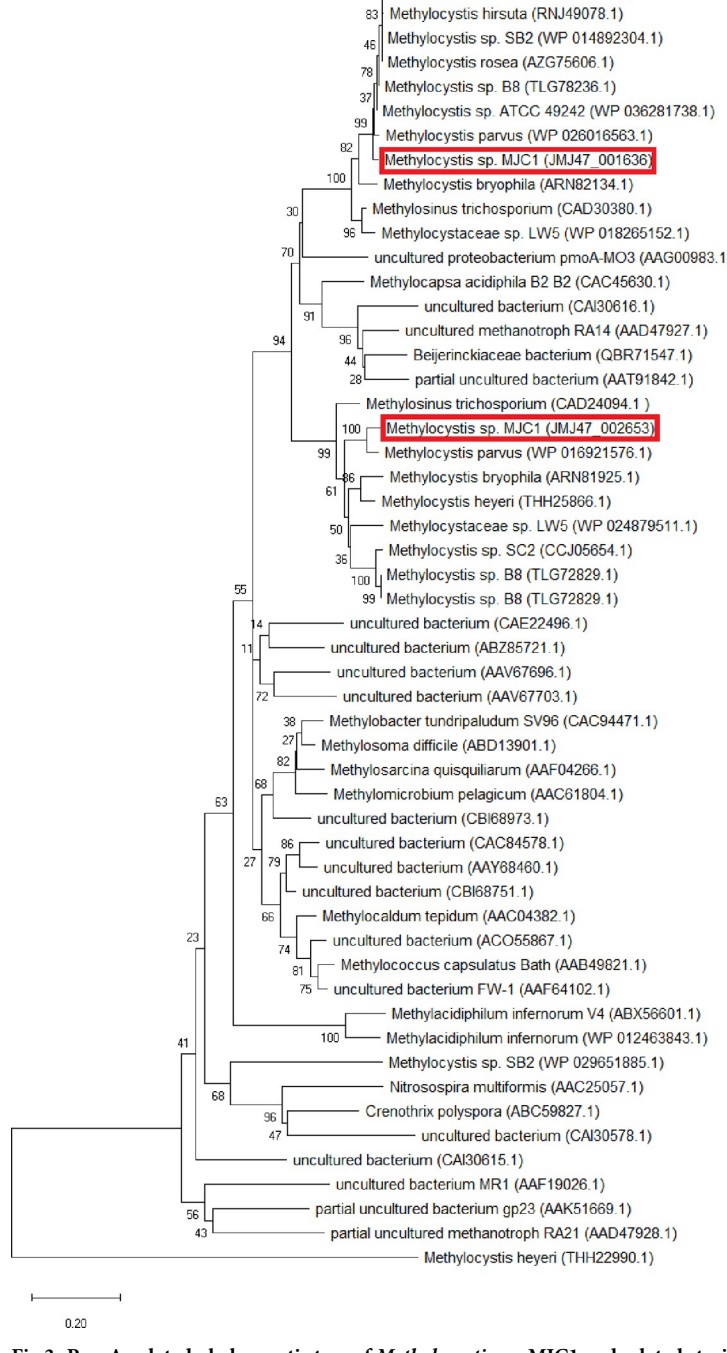

**Fig 3. PmoA-related phylogenetic tree of *Methylocystis* sp. MJC1 and related strains.** The related strains were selected as described in Jung et al [35]. The phylogenetic tree was reconstructed using neighbor-joining method. Values for bootstrap below 70% have been hidden. The bar in the lower left corner indicates 0.20 substitutions per amino acid site. The locations of pmoA1 (JMJ47_001636) and pmoA2 (JMJ47_002653) of *Methylocystis* sp. MJC1 are highlighted with a red box.

evolutionary closeness to *M. parvus* (**S1 Fig**). Notably, *Methylocystis* sp. MJC1 and *M. parvus* seem to be related, but distinct species, as electronic DDH did not confirm *M. parvus* to be the same species as *Methylocystis* sp. MJC1 (probability of DDH $\geq$ 70% is 0.01%). Furthermore, the values for both AAI (80.49%) and ANI (85%) fell short of the species boundary described in the literature, which is 95%. Therefore, *Methylocystis* sp. MJC1 may represent a distinct species within the genus *Methylocystis*.

## Functional genome annotation

Genome annotation suggests that *Methylocystis* sp. MJC1 has a functional particulate pMMO (**Fig 3** and **S3 Table**). pMMO is a membrane-bound enzyme that contains three subunits encoded by genes identified using the same names, viz. *pmoA*, *pmoB*, and *pmoC*. *pmoA* encodes the pMMO active site, while *pmoB* is responsible for the oxygenase activity. Together, these genes code for thirteen copper centers that are involved in the catalytic activity [43]. *Methylocystis* sp. MJC1 possesses the following two different pMMO-related operons in the genome: two copies of the *pmoCAB1* functional operon and one copy of the *pmoCAB2* functional operon. Furthermore, two singleton *pmoC1* paralogs and one singleton *pmoC2* paralog have been found in the genome. Some studies have shown that additional gene copies of *pmoC* are required for the growth of methanotrophs [44]. In addition, a previous study has hypothesized that *pmoC* paralogs could be responsible for methane sensing or gene expression, but not for catalytic activity [45]. The exact function of the singleton paralogs can be identified by mutational studies, if necessary.

*Methylocystis* sp. MJC1 has genes for sMMO based on genome annotation (**Table 1** and **S3 Table**). sMMO is a three-component protein complex that contains a hydroxylase (MMOH), reductase (MMOR), and regulatory component (MMOB) [46]. Each component is necessary for methane hydroxylation and NADH oxidation. MMOH is a dimer that contains α, β, and γ subunits and a hydroxy-bridged binuclear iron cluster. In diferrous state [$Fe^{II}$–$Fe^{II}$], this binuclear iron cluster can react with dioxygen and initiate methane hydroxylation [47]. The sMMO-related gene cluster encodes components in the following manner. MMOH is encoded by *mmoX*, *mmoY*, and *mmoZ*, while MMOR and MMOB are encoded by *mmoB* and *mmoC* genes respectively. The function of *mmoD* gene has not been reported yet; however, it has been hypothesized that *mmoD* might play an important role in binuclear iron cluster assembly [48]. The presence of sMMO may indicate increased metabolic versatility in comparison to other species that possess only one type of methane monooxygenase because of its ability to utilize methane in the presence or absence of copper in the medium [49]. Furthermore, sMMO has been shown to have a higher turnover number toward methane than pMMO, suggesting that switching to sMMO expression can lead to a higher growth rate when compared to other *Methylocystis* strains possessing only pMMO [50]. The growth rate of *Methylocystis* sp. MJC1 has been reported to be up to 0.063 $h^{-1}$ when copper was added to the medium [18]. Interestingly, the maximum growth rate (0.12 $h^{-1}$) was higher in the copper-less medium, indicating potentially higher growth rates in the presence of active sMMO. This value is higher than the growth rate observed for *Methylocystis* parvus (0.107 $h^{-1}$), which exhibits the highest among genus *Methylocystis*, based on available information [41]. One potential way to increase growth rate of *Methylocystis* sp. MJC1 is to express sMMO in high-pressure gas fermentation condition with a high methane mass.

Methanol dehydrogenase (MDH) is the second key methanotrophic enzyme that catalyzes the conversion of methanol into formaldehyde [51]. Methanotrophs possess two types of MDH depending on the ion type required for its activity: calcium-dependent (*mxaFI*) or lanthanide-dependent (*xoxF*) MDH. Although *xoxF* genes were discovered long ago, their

functions remained unknown until the recent discovery of lanthanides as co-factors. Almost all methanotrophs with calcium-dependent MDH also contain lanthanide-dependent MDH [52–54]. In the *Methylocystis* clade, all strains except KS32, have been reported to have both types of MDH [49]. The benefits of this ion flexibility are not yet understood. One explanation is that it can be a tool to control symbiotic metabolite transfer, such as methanol, to other members of the ecosystem. Another explanation is that it can be a form of adaptation in the case of changing environmental conditions.

*Methylocystis* sp. MJC1 was observed to contain genes for both enzymes (**Fig 3** and **S3 Table**). *mxaFI* is a soluble periplasmic pyrroloquinoline quinone-containing enzyme [51]. It is composed of two small and two large subunits that require calcium ions at the active site. MDH sends electrons to a specific variant of cytochrome cL (encoded by *mxaG*). The mxa operon also contains other genes that control calcium insertion and maturation of enzymes. The function of a gene, *moxJ*, which encodes a periplasmic solute-binding protein, has not been discovered yet. In contrast, lanthanide-dependent MDH has a less complex structure with only one large subunit present. One difference between the two enzymes is that lanthanide-dependent MDH has a distinguishable lanthanide-coordinating Asp residue located two positions away from the highly conserved Asp residue [51]. Investigations of the effects of lanthanide on methanol dehydrogenase activity have shown mixed results. In one study, *xoxF* activity was overruled by copper in *M. trichosporium* OB3b [53]. In another study, lanthanide-dependent MDH was found to be the preferred enzyme for methanol oxidation despite 100-fold higher calcium concentration with copper addition showing no effect on its activity [55]. Elucidation of the exact mechanism of the calcium-lanthanide switch, which appears to be present in *Methylocystis* sp. MJC1, may require additional experimental studies.

Genome analysis suggests that *Methylocystis* sp. MJC1 possesses tetrahydromethanopterin pathway (**Fig 4** and **S3 Table**). The tetrahydromethanopterin pathway is a linear pathway that participates in the oxidation of formaldehyde to formate under aerobic conditions. First discovered in methanogenic bacteria, this pathway is common in both methylotrophic and methanotrophic bacteria [56]. Formate formation is thought to be the key branch point for methane dissimilation and assimilation [57]. Genome analysis implies that in *Methylocystis* sp. MJC1, carbon assimilation occurs via the tetrahydrofolate pathway, which links formate to the serine cycle; dissimilation occurs by further oxidation of formate to carbon dioxide and subsequent release.

*Methylocystis* sp. MJC1 appears to have genes for a full serine cycle (**Table 1**, **Fig 4** and **S3 Table**). This is a characteristic of type II methanotrophs, as type I methanotrophs have only partial serine cycle. The serine cycle is the only oxygen-insensitive biochemical pathway that is capable of acetyl-CoA (C2 compound) synthesis from C1 compounds without any loss of carbon [58]. In addition, dissimilated carbon dioxide is assimilated back into the serine cycle, which allows for carbon recycling. In the serine cycle, glycine (C2 compound) generated from glyoxylate combines with methylene tetrahydrofolate (C1 compound) and generates serine (C3 compound), while the rest of the cycle regenerates the glycine acceptor [59].

Despite its metabolic versatility, the serine cycle is ATP-inefficient, which leads to low product and biomass yields in type II methanotrophs [60]. In contrast, the RuMP cycle, an assimilation pathway typically present in type I methanotrophs, supports 40–50% higher biomass yield [61]; this higher efficiency is attributed to lower ATP requirement. In RuMP cycle, formaldehyde (but not formate) reacts with ribulose monophosphate to produce hexulose phosphate, which is later converted into fructose-6-phosphate [62]. Pyruvate can then be produced via either Entner-Doudoroff (EDD) or glycolytic EMP pathway. Genome annotation implies that *Methylocystis* sp. MJC1 has a partial EMP variant of the RuMP pathway with only three missing genes: 6-phospho 3-hexuloisomerase (*phi*), 3-hexulose-6-phosphate synthase (*hps*), and

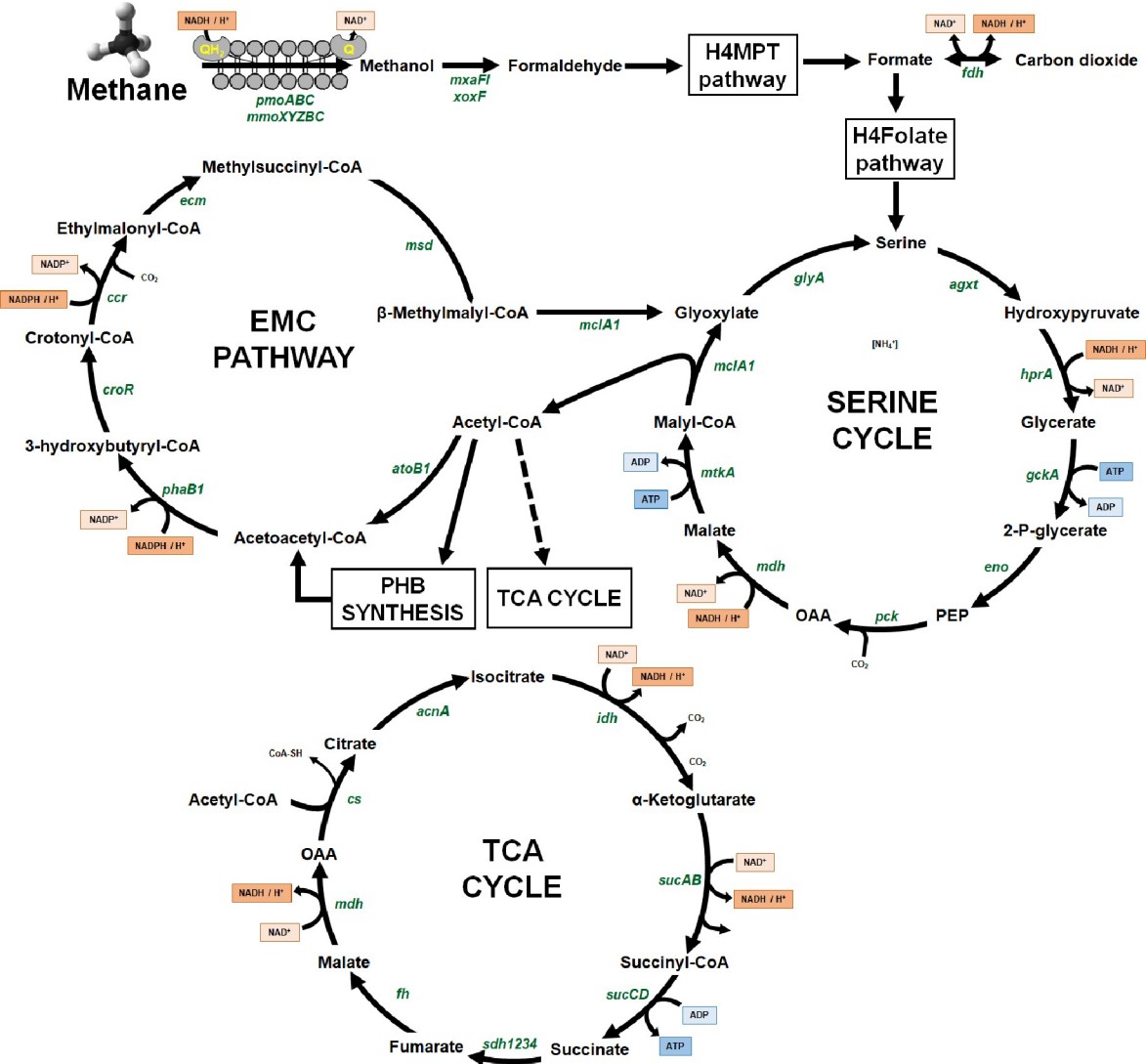

**Fig 4. Central carbon metabolism of *Methylocystis sp.* MJC1 based on functional genome annotation.** H$_4$MPT pathway refers to tetrahydromethanopterin pathway. H$_4$Folate pathway refers to tetrahydrofolate pathway. The name of the genes involved in the reactions are depicted in green color and correspond to gene names provided in S3 Table. The names of proteins encoded by the genes are also provided in S3 Table.

transaldolase (*tal*). This potentially opens an opportunity for a metabolic engineering strategy to recreate the full RuMP pathway to achieve higher yields, similar to the way already achieved in *E. coli* [63].

The serine cycle requires constant glyoxylate regeneration. Since methanotrophs usually lack the glyoxylate shunt, they have evolved another pathway for glyoxylate regeneration [64]. *Methylocystis* sp. MJC1 has genes for a fully functional ethylmalonyl-CoA (EMC) pathway, a representative type II methanotroph pathway for glyoxylate regeneration [45, 62]. The EMC pathway branches off of the serine cycle via malyl-CoA lyase, which produces acetyl-CoA and regenerates glyoxylate at the same time. Furthermore, certain organic compounds, such as esters, alcohols, and fatty acids, require the presence of acetyl-CoA for their metabolization. The EMC pathway may serve as an option for converting acetyl-CoA (2C compound) into relevant 4C compounds. One interesting compound that can be produced from EMC pathway

branching is PHB. PHB is a biodegradable plastic polymer that is used as an energy source and carbon sink under stress condition of nitrogen depletion [41]. This fact, coupled with atmospheric methane elimination, makes type II methanotrophs a promising solution for solving environmental pollution problems. A related species, *M. hirsuita*, has been shown to accumulate PHB over 40% of its dry weight [65]. Studies with other type II methanotroph species have shown promising results for PHB production [40, 65, 66].

Both the EMC pathway and serine cycle have common reactions with Krebs cycle, thereby creating interconnected metabolic cycles. Krebs cycle contains malate dehydrogenase that is shared with the serine cycle, and fumarase and succinate dehydrogenase that are shared with the EMC pathway. This unusual network organization allows for C2 and C4 intermediate interplay between different pathways to maintain the carbon flux balance [67]. Organism-specific genome-scale models have been applied to predict such flux distributions in type II methanotrophs [65–66]. The sequenced genome of *Methylocystis* sp. MJC1 may assist in the potential development of an organism-specific genome-scale model.

Regarding nitrogen metabolism, both experimental evidence and genome annotation suggest that *Methylocystis* sp. MJC1 is capable of growing on nitrate- and ammonium-containing substrates (S3 Table). Thus, either nitrate or ammonium-based mineral slats media can be used for large-scale culture of *Methylocystis* sp. MJC1 and production of biodegradable polyhydroxyalkanoates. The oxidation state of nitrogen in nitrate is higher than that in amino acids. Therefore, nitrate must be reduced to ammonium in order to be assimilated. This comes at the expense of a reduced electron pool. Ammonium can act as both a nitrogen source and a competitive inhibitor of methane monooxygenase due to its homology with ammonia monooxygenase (AMO) [68]. The result of ammonia oxidation is the production of hydroxylamine, a toxic chemical compound. Methanotrophs have developed two distinct mechanisms to prevent excessive hydroxylamine accumulation. In one mechanism, which is similar to that observed for ammonia oxidizers, the resultant hydroxylamine is oxidized to nitrite by hydroxylamine oxidoreductase, and then nitrite is converted to nitric oxide, followed by further reduction to nitrous oxide [68]. In another completely distinct mechanism, hydroxylamine is converted to ammonium via hydroxylamine reductase. Regardless of how hydroxylamine detoxification is achieved, ammonia assimilation occurs at the expense of reducing the electron demand [68]. *Methylocystis* sp. MJC1 has genes for both the pathways for hydroxylamine detoxification. Whether the growth rate is higher in the presence of ammonium than that in the presence of nitrate depends on whether the electron demand for hydroxylamine detoxification outweighs the electron demand for nitrate reduction. In our experiments, *Methylocystis* sp. MJC1 was capable to grow using either nitrate or ammonium as nitrogen source, a common characteristic among methanotrophs (S3 Table). Other type II methanotrophs, including several other *Methylocystis* species, have been reported to have higher growth rate in the presence of ammonia [65, 66].

*Methylocystis* sp. MJC1's genome has genes for two routes for ammonia assimilation: glutamine synthetase/glutamate synthetase (GS/GOGAT) pathway and glutamate dehydrogenase pathway (S3 Table). GS/GOGAT is a high-affinity pathway that is activated under low ammonia concentrations. In this pathway, glutamine synthetase catalyzes glutamine formation from ammonia and glutamate, while glutamate synthase transaminates 2-oxoglutarate and glutamine to regenerate glutamate, resulting in a net gain of one glutamate molecule [69]. On the other hand, glutamate dehydrogenase is a low-affinity pathway that is activated under high ammonia concentrations. In this pathway, reductive amination of 2-oxoglutarate occurs [70]. Other type II methanotrophs, such as *M. trichosporium* OB3b, are known to assimilate ammonium exclusively via the GS-GOGAT pathway [66]. Therefore, it is expected that *Methylocystis* sp. MJC1 must also be utilizing the GS/GOGAT pathway for ammonia assimilation.

**Table 2. COG functional classification of *Methylocystis* sp. MJC1 genome.** The names of super categories are indicated in bold uppercase letters.

| Category | Functional Classification | Number of genes |
|---|---|---|
| **CELLULAR PROCESSES AND SIGNALING** | | |
| D | Cell cycle control, cell division, and chromosome partitioning | 35 |
| M | Cell wall/membrane/envelope biogenesis | 250 |
| N | Cell motility | 63 |
| O | Post-translational modification, protein turnover, and chaperones | 153 |
| T | Signal transduction mechanisms | 255 |
| U | Intracellular trafficking, secretion, and vesicular transport | 10 |
| V | Defense mechanisms | 53 |
| W | Extracellular structures | 0 |
| Y | Nuclear structure | 0 |
| Z | Cytoskeleton | 0 |
| **INFORMATION STORAGE AND PROCESSING** | | |
| A | RNA processing and modification | 0 |
| B | Chromatin structure and dynamics | 2 |
| J | Translation, ribosomal structure, and biogenesis | 181 |
| K | Transcription | 206 |
| L | Replication, recombination, and repair | 236 |
| **METABOLISM** | | |
| C | Energy production and conversion | 275 |
| E | Amino acid transport and metabolism | 224 |
| F | Nucleotide transport and metabolism | 67 |
| G | Carbohydrate transport and metabolism | 120 |
| H | Coenzyme transport and metabolism | 166 |
| I | Lipid transport and metabolism | 150 |
| P | Inorganic ion transport and metabolism | 213 |
| Q | Secondary metabolites biosynthesis, transport, and catabolism | 69 |
| **POORLY CHARACTERIZED** | | |
| R | General function prediction only | 340 |
| S | Function unknown | 379 |

## Cluster of orthologous genes and pangenome analysis

Cluster of orthologous genes (COG) is an attempt to functionally classify genomes. In other words, COG provides an overview of gene functions in an organism of interest [71]. The most common COG category in *Methylocystis* sp. MJC1 was observed to be an unknown function with 379 genes. Among the genes that were categorized into functional categories, the top three categories were found to be general function prediction only (340 genes), energy production and conversion (275 of all genes), and signal transduction mechanisms (255 genes) (**Table 2**).

In addition to COG, a pan-genome analysis was performed to identify *Methylocystis* sp. MJC1 COG in the context of other *Methylocystis* species strains. The pan-genome analysis showed that *Methylocystis* sp. MJC1 shares 1322 core genes and 1833 accessory genes with other *Methylocystis* species strains (**S4 Table**). In addition, it contains 739 unique genes that are present only in *Methylocystis* sp. MJC1 and 208 exclusively absent genes, that is, genes that are absent only in *Methylocystis* sp. MJC1 (**S5 Table**).

## Conclusion

The genomic properties of a promising PHB-producing type II methanotroph strain, *Methylocystis* sp. MJC1, were analyzed. The genome of *Methylocystis* sp. MJC1 has been found to contain one chromosome and two plasmids. Comparative genomics reconfirmed the previous suggestion that *Methylocystis* sp. MJC1 may represent a distinct species. Pan-genome analysis implied that the strain may have different transcription regulatory mechanisms and less metabolic flexibility compared to other *Methylocystis* strains. Functional genome annotation has shown that the strain contains the EMC and serine cycle, which share reactions with Krebs cycle. This interconnected network organization could be further studied using strain-specific genome-scale model reconstruction. Interestingly, the strain was observed to possess both particulate and soluble methane monooxygenases, which might be related to its higher metabolic flexibility and growth rate than other *Methylocystis* strains.

## Supporting information

**S1 Table. Average Amino acid Identity (AAI) comparison between *Methylocystis* sp. MJC1 and *Methylocystis* genus clade.** AAI values are provided in percentage. *Methylocystis* sp. MJC1 had total 4306 genes identified during AAI.
(DOCX)

**S2 Table. Average Nucleotide Identity (ANI) comparison between *Methylocystis* sp. MJC1 and *Methylocystis* genus clade.** ANI values are provided in percentage.
(DOCX)

**S3 Table. Central carbon metabolism-related genes in *Methylocystis* sp. MJC1.** The names of major biochemical pathways or enzymes are written in bold.
(DOCX)

**S4 Table. Cluster of orthologous genes (COG) functional classification for unique genes identified in *Methylocystis* sp. MJC1 by pangenome analysis.** The names of super categories are written in bold uppercase letters.
(DOCX)

**S5 Table. COG functional classification for exclusively absent genes identified in *Methylocystis* sp. MJC1 by pangenome analysis.** The names of super categories are written in bold uppercase letters.
(DOCX)

**S1 Fig. Concatenated core gene (core genome) phylogenic tree for *Methylocystis* species strains.** The location of core genes for *Methylocystis* sp. MJC1 is highlighted in red box. MYA represents evolutionary age of genes in million years ago.
(DOCX)

## Author Contributions

**Formal analysis:** Sanzhar Naizabekov, Seung Woon Hyun.

**Funding acquisition:** Ok Kyung Lee, Eun Yeol Lee.

**Investigation:** Sanzhar Naizabekov, Seung Woon Hyun, Jeong-Geol Na.

**Resources:** Sukhwan Yoon.

**Supervision:** Ok Kyung Lee, Eun Yeol Lee.

**Writing – original draft:** Sanzhar Naizabekov, Seung Woon Hyun.

**Writing – review & editing:** Ok Kyung Lee, Eun Yeol Lee.

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
