## [Decision Letter · Decision Letter 0]

30 Jan 2023

PONE-D-23-00366Dear Editors,

Full title is, "Comparative genomic analysis of Methylocystis sp. MJC1 as a platform strain for polyhydroxybutyrate biosynthesis"PLOS ONE

Dear Dr. Eun Yeol Lee,

Thank you for submitting your manuscript to PLOS ONE. After careful consideration, we feel that it has merit but does not fully meet PLOS ONE’s publication criteria as it currently stands. Therefore, we invite you to submit a revised version of the manuscript that addresses the points raised during the review process.Please revised the manuscript carefully by addressing the concerns raised by the two reviewers (see below)  before acceptance for publication accordingly. In addition, to improve the quality of the introduction,  include minor modifications in each paragraph as follows:Paragraph 1: State the problemParagraph 2: State what is known and what is unknownParagraph 3: State your aim and hypothesisParagraph 4: State what the current study inform other studiesPlease submit your revised manuscript by **Mar 16, 2023**. If you will need more time than this to complete your revisions, please reply to this message or contact the journal office at plosone@plos.org. Please include the following items when submitting your revised manuscript:A rebuttal letter that responds to each point raised by the academic editor and reviewer(s). You should upload this letter as a separate file labeled 'Response to Reviewers'.A marked-up copy of your manuscript that highlights changes made to the original version. You should upload this as a separate file labeled 'Revised Manuscript with Track Changes'.An unmarked version of your revised paper without tracked changes. You should upload this as a separate file labeled 'Manuscript'.If applicable, we recommend that you deposit your laboratory protocols in protocols.io to enhance the reproducibility of your results. Protocols.io assigns your protocol its own identifier (DOI) so that it can be cited independently in the future. For instructions see: https://journals.plos.org/plosone/s/submission-guidelines#loc-laboratory-protocols. Additionally, PLOS ONE offers an option for publishing peer-reviewed Lab Protocol articles, which describe protocols hosted on protocols.io. Read more information on sharing protocols at https://plos.org/protocols?utm_medium=editorial-email&utm_source=authorletters&utm_campaign=protocols.

We look forward to receiving your revised manuscript.

Kind regards,

Bashir Sajo Mienda, PhD

Academic Editor

PLOS ONE

“This research was supported by the C1 Gas Refinery Program through the National Research Foundation of Korea (NRF) funded by the Ministry of Science and ICT (2015M3D3A1A01064882 and 2015M3D3A1A01064926). This research was also supported by Basic Science Research Program through the National Research Foundation Korea (NRF) funded by the Ministry of Education (2020R1I1A1A01073467).”

Reviewers' comments:

Reviewer's Responses to Questions

**Comments to the Author**

1. Is the manuscript technically sound, and do the data support the conclusions?

Reviewer #1: Yes

Reviewer #2: Yes

2. Has the statistical analysis been performed appropriately and rigorously? 

Reviewer #1: N/A

Reviewer #2: No

3. Have the authors made all data underlying the findings in their manuscript fully available?

Reviewer #1: Yes

Reviewer #2: Yes

4. Is the manuscript presented in an intelligible fashion and written in standard English?

Reviewer #1: Yes

Reviewer #2: Yes

5. Review Comments to the Author

Reviewer #1: I have read the manuscript entitled “Comparative genomic analysis of Methylocystis sp. MJC1 as a platform strain for polyhydroxybutyrate biosynthesis” authored by Naizabekov et al. The authors performed genomic analysis of Methylocytis sp. MJC1, which can utilize methane and produce PHB, a promising bioplastic with industrial interests. Although methane utilization gets more important, information about methanotrophs is limited compared to other conventional hosts. In general, I think this this manuscript meets the readership of Plos One. However, I found several issues that need to be addressed its publication. My specific comments are below:

Major points

1. The authors should carefully cite recent reference(s) which can directly support a statement in the manuscript. For example, in lines 55-57, Ref 4 does not seem appropriate to support “The maximum carbon conversion efficiency of methane to methanol oxidation using chemical processes has been reported to be below 50%, while that of methanotrophs was reported to be up to 75% [4].”

2. The introduction section needs to be improved.

- The introduction section lacks general information about methanotrophs and did not introduce why more studies for methylocytis species are needed.

- The authors may want to include general descriptions about multiple methanotroph genus with elaborating on how they are different (lines 63-71). This information might help to understand the novelty Methylocytis sp. MJC1 if it is. Lines 70-71, one might ask why we need more strains?

- Please explain how different type I and type II methanotrophs are.

3. Regarding the genome analysis,

- Lines 109-118, generally, PacBio reads are utilized first to draft a genome and then the draft genome is further refined by using illumina reads. If this is the case, the authors may want to correct the current text. Otherwise, please elaborate.

- How was the 4,538,495 bp calculated to be the total genome? A sum of the lengths of the chromosome and the two plasmids is not equal to this number.

- What does “status” in Table 1 means?

4. Lines 149-152, was the faster growth rate due to the change in the medium composition? If I read it correctly, the change resulted in a 20X faster growth on methane. Are the two compared strains identical? The authors may want to compare growth rates of methanotrophs side by side.

5. If the fact that this strain has both pMMO and sMMO differentiate it from other methanotrophs, it would be nice to check the expression of the genes encoding them in a methane utilizing condition.

Minor points

1. Lines 68-69, “One practical ~ target products”. The authors did not mention and compare PHB productivity of methanotrophs. This statement should be re-written or clarified.

2. Lines 171-172, unless the authors meant that it has two linear chromosomes, I think “The chromosome has gaps and consists of two linear contigs,” should be revised with replacing “the chromosome” with “the assembled genomes”.

3. Not all gene names were italicized.

4. Regarding Figure 2, the authors should detail the meaning of each color in the legend.

5. Line 214, Fe(II)

Reviewer #2: The manuscript presents Methylocystis sp. MJC1 as a new potential platform for the production of biodegradable polyhydroxybutyrate (PHB) from methane. Upon comparison with other Methylocystis species, Methylocystis sp. MJC1 appears more performant in this task. Beside the analysis of PHB production, a comparative full-genome analysis was carried out to underline similarities and differences with other Methylocystis species. An interesting result is a possible engineering strategy to improve the biomass yield.

Overall, the manuscript is well-written and interesting. I have two main comments only:

- would it be possible to associate the growth rates with an error and carry out a statistical comparison?

- In the "Conclusion" I would suggest to emphasize the main result, i.e., Methylocystis sp. MJC1 is a new potential platform for PHB production.

Minor comments:

- line 147: a blank space is missing between "MJC1" and "as";

- line 150: higher "than" rather than "to the";

- Table : "PBH synthesis gene", why nothing is written in columns 3-5?

6. PLOS authors have the option to publish the peer review history of their article (what does this mean?). If published, this will include your full peer review and any attached files.

Reviewer #1: No

Reviewer #2: No

---

## [Author Response · Author response to Decision Letter 0]

4 Apr 2023

Responses to Reviewers’ Comments

We appreciate the Reviewers for their invaluable comments. As explained below, we have revised our original submission in response to all the Reviewer’s comments. What follows are our point-by-point responses to the reviewer’s comments.

Reviewer #1: I have read the manuscript entitled “Comparative genomic analysis of Methylocystis sp. MJC1 as a platform strain for polyhydroxybutyrate biosynthesis” authored by Naizabekov et al. The authors performed genomic analysis of Methylocystis sp. MJC1, which can utilize methane and produce PHB, a promising bioplastic with industrial interests. Although methane utilization gets more important, information about methanotrophs is limited compared to other conventional hosts. In general, I think this this manuscript meets the readership of Plos One. However, I found several issues that need to be addressed its publication. My specific comments are below:

Major points

1. The authors should carefully cite recent reference(s) which can directly support a statement in the manuscript. For example, in lines 55-57, Ref 4 does not seem appropriate to support “The maximum carbon conversion efficiency of methane to methanol oxidation using chemical processes has been reported to be below 50%, while that of methanotrophs was reported to be up to 75% [4].”

(Response) We have rechecked all the references. We think the references 4 cited in the main text are appropriate.

2. The introduction section needs to be improved.

- The introduction section lacks general information about methanotrophs and did not introduce why more studies for Methylocystis species are needed.

- The authors may want to include general descriptions about multiple methanotroph genus with elaborating on how they are different (lines 63-71). This information might help to understand the novelty Methylocystis sp. MJC1 if it is. Lines 70-71, one might ask why we need more strains?

- Please explain how different type I and type II methanotrophs are.

(Response) According to the reviewer's comments, we have revised the introduction as follows:

Line 59-84, 

Methanotrophs are traditionally categorized into two groups based on their carbon assimilation pathways: Type I and Type X belong to gamma-proteobacteria, while Type II belongs to alpha-proteobacteria. Type I and X use the ribulose monophosphate (RuMP) cycle to assimilate C1 carbon sources such as formaldehyde, whereas Type II uses the serine cycle to assimilate formate generated by methane oxidation [5]. 

Type I methanotrophs including Methylomonas sp. and Methylomicrobium sp. condense formaldehyde to ribulose monophosphate, producing fructose-6-phosphate. This metabolite is integrated into various metabolic pathways such as the Embden-Meyerhof-Parnas, oxidative, and non-oxidative pentose phosphate pathways [6]. In order to produce high-value products such as 2,3-BDO, succinic acid, shinorine, and auxin from methane, native or non-native biosynthetic pathway have been successfully introduced into central carbon metabolism of type I methanotrophs[7-10]. Nonetheless, the low productivity of these target products needs to be addressed. 

Type II methanotrophs, such as Methylocystis sp. and Methylosinus sp., have the ability to accumulate polyhydroxybutyrate (PHB) around 34-70% of dry cell weight through a series of enzymatic reactions involving acetyl-CoA acetyltransferase, acetoacetyl-CoA reductase, and PHA synthase [11-13]. Recently, PHB production by Type I methanotroph Methylomicrobium alcaliphilum 20Z has also been reported, but its feasibility of commercialization remains low [14]. Therefore, Type II methanotrophs are considered promising PHB platform strains. In particular, Methylocystis sp. has even more potential because of its higher growth rate (up to 0.12-0.16 h–1) compared to Methylosinus sp. [15]. Despite these promising advantages, only six species of the Methylocystis genus have been reported, thus more strain needs to be isolated and characterized to develop PHB-producing platforms [16-17]. 

3. Regarding the genome analysis,

- Lines 109-118, generally, PacBio reads are utilized first to draft a genome and then the draft genome is further refined by using illumina reads. If this is the case, the authors may want to correct the current text. Otherwise, please elaborate.

(Response) We agree with the reviewer's comments. We have revised the sentence as follows:

Lines 124-127,

The complete genome sequencing of Methylocystis sp. MJC1 was performed using Macrogen's PacBio RS II and Illumina HiSeq 2500 sequencing platforms. PacBio reads were used to generate a genome draft, which was then further refined using Illumina reads.

- How was the 4,538,495 bp calculated to be the total genome? A sum of the lengths of the chromosome and the two plasmids is not equal to this number.

(Response) We have removed incorrect information from the manuscript.

- What does “status” in Table 1 means?

(Response) The 'status' in Table 1 means 'Assembly level'. We have changed 'status' to 'Assembly level' and modified Table 1. 

4. Lines 149-152, was the faster growth rate due to the change in the medium composition? If I read it correctly, the change resulted in a 20X faster growth on methane. Are the two compared strains identical? The authors may want to compare growth rates of methanotrophs side by side. 

(Response) The growth rates were different because the culture conditions were quite different. Specifically, in this study, the strain was cultured using NMS medium containing 1g KNO3/L in a bioreactor system that was supplied with methane and air in an optimal ratio, and the stirring speed was adjusted according to cell growth. In contrast, in the previous paper, cell culture was performed based on DNMS medium with nitrogen levels reduced down to 1/5, and conducted in flasks where methane or oxygen might be depleted during the culture process. Insufficient supply of methane, oxygen and nitrogen source can slow down the cell growth. Thus, there were differences in growth rates. 

5. If the fact that this strain has both pMMO and sMMO differentiate it from other methanotrophs, it would be nice to check the expression of the genes encoding them in a methane utilizing condition.

(Response) Although Methylocystis parvus OBBP, a well-known Methylocystis species, does not contain sMMO, some Methylocystis sp. Strains possess sMMO. Generally, it is known that pMMO is expressed under copper-rich conditions (> 5uM), while sMMO is expressed only under low-copper conditions. Currently, we are conducting and optimizing gas fermentation to obtain high cell density and enhance PHB production. The expression of each gene will be analyzed using qPCR or SDS-PAGE. 

Minor points

1. Lines 68-69, “One practical ~ target products”. The authors did not mention and compare PHB productivity of methanotrophs. This statement should be re-written or clarified.

(Response) We have revised the introduction based on the reviewer’s comments.

2. Lines 171-172, unless the authors meant that it has two linear chromosomes, I think “The chromosome has gaps and consists of two linear contigs,” should be revised with replacing “the chromosome” with “the assembled genomes”.

(Response) We have revised the sentence based on the reviewer's comment.

Line 171-172, 

“The assembled genomes has gaps and consists of two linear contigs, whereas both plasmids are complete and circular.”

3. Not all gene names were italicized.

(Response) We have reviewed and changed them.

4. Regarding Figure 2, the authors should detail the meaning of each color in the legend. 

(Response) We have added the following description for Figure 2.

Figure 2. Methylocystis sp. MJC1 contigs in circular representation. This is a visual representation of the circular map of the MJC1 genome. The genome map consists of three circles. The two outer circles display the open reading frame, while the inner circle represents the GC skew. Positive and negative values of the GC skew are depicted by green and purple, respectively. The GC content is displayed in black. This genome map was generated using the CGView Server.

5. Line 214, Fe(II)

(Response) We have changed ' In different states [FeII−FeII]' to ' In diferrous state [FeII−FeII]' in line 214.

Reviewer #2: The manuscript presents Methylocystis sp. MJC1 as a new potential platform for the production of biodegradable polyhydroxybutyrate (PHB) from methane. Upon comparison with other Methylocystis species, Methylocystis sp. MJC1 appears more performant in this task. Beside the analysis of PHB production, a comparative full-genome analysis was carried out to underline similarities and differences with other Methylocystis species. An interesting result is a possible engineering strategy to improve the biomass yield.

Overall, the manuscript is well-written and interesting. I have two main comments only:

- would it be possible to associate the growth rates with an error and carry out a statistical comparison?

- In the "Conclusion" I would suggest to emphasize the main result, i.e., Methylocystis sp. MJC1 is a new potential platform for PHB production.

Minor comments:

- line 147: a blank space is missing between "MJC1" and "as";

(Response) The sentence has been revised as suggested by the reviewer

- line 150: higher "than" rather than "to the";

(Response) We have revised that sentence as follows:

The maximum growth rate of the Methylocystis sp. MJC1 in the exponential growth phase was 0.12 h-1, which is higher than the previously reported cell growth rate of MJC1 (0.063 h-1) cultured in DNMS (dilute nitrate mineral salts) medium with the addition of copper.

- Table : "PHB synthesis gene", why nothing is written in columns 3-5?

(Response) We presented the presence or absence of PHB synthesis genes in the table 1.

---

## [Editor Report · Decision Letter 1]

10 Apr 2023

Comparative genomic analysis of Methylocystis sp. MJC1 as a platform strain for polyhydroxybutyrate biosynthesis

PONE-D-23-00366R1

Dear Dr. LEE,

We’re pleased to inform you that your manuscript has been judged scientifically suitable for publication and will be formally accepted for publication once it meets all outstanding technical requirements.

Kind regards,

Bashir Sajo Mienda, PhD

Academic Editor

PLOS ONE
---

## [Editor Report · Acceptance letter]

19 Apr 2023

PONE-D-23-00366R1 

Comparative genomic analysis of *Methylocystis* sp. MJC1 as a platform strain for polyhydroxybutyrate biosynthesis 

Dear Dr. Lee:

I'm pleased to inform you that your manuscript has been deemed suitable for publication in PLOS ONE. Congratulations! Your manuscript is now with our production department. 

Kind regards, 

on behalf of

Dr. Bashir Sajo Mienda 

Academic Editor

PLOS ONE